# Prevalence of burnout syndrome in Brazilian anesthesiologists during the COVID-19 pandemic: A cross-sectional survey

Liana M. T. A. Azi[1]*, Thaiane S. Ferreira[2], Thiago Cerqueira-Silva[3], Luis A. S. Diego[4], Marcos A. C. Albuquerque[5], Matheus L. Azi[6]

1 Department of Anesthesiology and Surgery, Professor Edgard Santos University Hospital, Federal University of Bahia, Salvador, Bahia, Brazil, 2 Faculty of Medicine, Federal University of Bahia, Salvador, Bahia, Brazil, 3 Gonçalo Moniz Institute, Oswaldo Cruz Foundation, Salvador, Bahia, Brazil, 4 Department of General and Specialized Surgery, Federal Fluminense University, Rio de Janeiro, Rio de Janeiro, Brazil, 5 Department of Medicine, Federal University of Sergipe, Aracaju, Sergipe, Brazil, 6 Manoel Victorino Hospital, Secretary of Health for the State of Bahia, Salvador, Bahia, Brazil

* liana.araujo@ufba.br

**Data Availability Statement:** All relevant data are within the paper and its Supporting Information files.

## Abstract

### Background

Burnout syndrome, one of the consequences of chronic exposure to stressful, is more prevalent among physicians compared to the general population. Anesthesiology, alongside high-stress specialties such as emergency medicine and surgery, is particularly susceptible to this condition. During the COVID-19 pandemic, anesthesiologists were often on the front lines, potentially exacerbating burnout. This study aimed to assess the prevalence of burnout syndrome among Brazilian anesthesiologists during the pandemic.

### Methods

A cross-sectional analytical observational study was conducted with all members of the Brazilian Society of Anesthesiology (SBA). Data were collected via sociodemographic questionnaires and the Maslach Burnout Inventory (MBI), disseminated by email.

### Results

Burnout syndrome was identified in 19.6% (n = 213) of respondents, while 56.5% (n = 613) were at high risk for developing burnout. Having considered quitting the specialty was the variable most strongly associated with the prevalence of burnout syndrome and the high risk of burnout. As a protective factor, dedicating more time to leisure (over 5 hours per week) was related to a lower occurrence of burnout syndrome and its risk.

### Conclusion

Burnout syndrome is highly prevalent among Brazilian anesthesiologists and residents. Target strategies to mitigate burnout should be implemented by healthcare institutions, professional organizations, and government bodies.

**Funding:** The author(s) received no specific funding for this work.

**Competing interests:** NO authors have competing interests.

## Introduction

Burnout syndrome, defined as a state of physical and mental exhaustion primarily caused by professional life, was first described as "staff burnout" by Freudenberger in 1974 [1]. Its occurrence is associated with the individual's inadequate and erratic adaptation to daily work dynamics, leading to the chronic manifestation of occupational stress [2]. The current definition was proposed by Maslach and Jackson [3], who characterized it as a syndrome comprising three dimensions: emotional exhaustion (EE), depersonalization (DE), and reduced personal accomplishment (RRP).

Maslach and Leiter state that "due to the nature and functionality of the position, there are professions at risk and high risk" for the development of the syndrome [2]. Medicine falls into the category of high-risk professions, and anesthesiology, in particular, stands out for its ability to promote high stress levels [4]. The critical nature of their work, which involves managing life-threatening situations and ensuring patient safety during surgery, contributes significantly to this stress. Anesthesiologists often work long hours with irregular schedules, face high demands for precision and rapid decision-making, and manage complex patient cases, all of which add to the unique stressors of the specialty [5].

Burnout appears to arise from an imbalance between personal and situational factors [6]. Emotional exhaustion (EE) refers to the feeling of emotional and physical depletion. It is the recognition that one no longer possesses the energy to sustain work activities. The work routine becomes burdensome and distressing. Depersonalization (DE) manifests through emotional detachment from the individuals the professional should serve and their colleagues. Interactions become impersonal and devoid of empathy, sometimes accompanied by cynical or sarcastic behavior. DE acts as a defensive mechanism within the syndrome. Reduced personal accomplishment (RRP) occurs when an individual loses satisfaction and efficacy in their work. They experience a sense of personal discontent, and work transforms into a burden [2].

The COVID-19 pandemic has significantly impacted healthcare systems worldwide, leading to unprecedented levels of stress among healthcare professionals, including anesthesiologists [7]. Increased workload, risk of infection, and changes in work protocols have all contributed to heightened stress levels [8].

Burnout among anesthesiologists became an even more critical issue after the COVID-19 pandemic, with international studies reporting prevalence rates ranging from 30% to 60% [9–11]. In Brazil, pre-pandemic data on anesthesiologists' burnout prevalence was limited, with rates ranging from 2.43% to 10.4% in small, localized samples [12, 13]. The increased stressors introduced by the COVID-19 pandemic have likely exacerbated this situation.

Given this context, this study aimed to identify the prevalence of burnout syndrome or high risk for burnout among anesthesiologists who are members of the Brazilian Society of Anesthesiology (SBA) during the COVID-19 pandemic. We hypothesize that these increased stressors may have led to a higher prevalence of burnout among anesthesiologists during the pandemic compared to pre-pandemic periods.

## Methods

### Study design

This cross-sectional survey aimed to determine the prevalence of burnout syndrome or high risk for burnout among anesthesiologists who are members of the Brazilian Society of Anesthesiology (SBA) during the COVID-19 pandemic. The study adhered to the Strengthening the Reporting of Observational Studies in Epidemiology (STROBE) guidelines [14] for reporting observational research, and ethical approval was obtained (number 4.098.345) from the Brazilian National Platform.

## Participants

All members of the SBA, anesthesiology physicians or residents, were invited to participate in the survey through SBA e-mails. Participation was voluntary, and the responses were anonymous. Prior to accessing the questionnaire, participants were required to provide written informed consent.

## Ethical considerations

Informed consent was obtained from all participants before they accessed the survey. The consent form explained the study's purpose, procedures, potential risks, and benefits. It also assured participants that their responses would be confidential and anonymized, and that they could withdraw from the study at any time without any repercussions. Data confidentiality was strictly maintained by using an anonymous online platform that prevented multiple responses from the same participant and did not collect any identifying information. The sociodemographic characteristics of the SBA members were provided to the researchers to determine the sample's representativeness, ensuring that no individual participant could be identified.

## Sampling method

The sample size was determined based on the total number of complete survey responses received, as no statistical power calculation was conducted prior to the study. This approach was chosen due to the exploratory nature of the research and the aim to capture a comprehensive snapshot of the entire population of Brazilian anesthesiologists during the pandemic.

## Participant recruitment

The recruitment of participants was carried out by sending invitations on three specific dates: 21 June 2021, 17 January 2022, and 16 September 2022. These dates were selected based on the availability of the mailing list from the Brazilian Society of Anesthesiology, ensuring broad and representative participation.

## Survey questionnaire

Participants first completed a questionnaire that included 20 items covering demographic information, work routines, leisure time, and a specific question about the influence of the pandemic on their responses (S1 File). Burnout was assessed using the validated Maslach Burnout Inventory Human Services Survey (MBI) (3), which consists of 22 questions assessing the three dimensions of burnout: emotional exhaustion (9 questions), depersonalization (5 questions), and personal accomplishment (8 questions). It is a widely used instrument for assessing burnout, with extensive validation across various populations, including healthcare professionals and it utilizes a 7-point Likert scale, ranging from 'never' to 'every day', to capture the frequency of experiences. The reliability of the MBI has been demonstrated in numerous studies, with Cronbach's alpha coefficients typically ranging from 0.70 to 0.90 for the subscales. The validity of the MBI is supported by its strong correlations with related constructs such as job satisfaction, organizational commitment, and mental health outcomes. Cut-off scores were used to characterize burnout syndrome: $\geq 27$ points for emotional exhaustion, $\geq 10$ points for depersonalization, and $\leq 33$ points for reduced personal accomplishment.

High risk for burnout was identified based on a previous study, that classifies it as the combination of high scores on emotional exhaustion ($\geq 27$) and/or depersonalization ($\geq 10$) [15].

## Data collection

Data were collected using an online platform (https://pt.surveymonkey.com/r/burnout_anestesiologia) without participant identification. The system was programmed to prevent multiple responses from the same participant.

Questionnaires missing any response of the MBI questions or with less than 80% of the demographic questions were excluded.

The SBA provided researchers with the sociodemographic characteristics of its members to determine the sample's representativeness.

## Statistical analysis and variable selection for adjusted models

All statistical analyses were performed using R software, version 4.0.2. Categorical variables were analyzed using Pearson's $\chi2$ test and presented as absolute and relative frequencies. Continuous variables were described as medians and interquartile ranges.

For the outcomes of burnout and high risk for burnout, we performed univariable analyses using the $\chi2$ test. Variables with a possible association to either outcome ($p < 0.1$ in univariable analyses) were included in separate multivariable models for burnout and high risk for burnout. In addition to the variables with $p<0.1$, we also included age and sex in all models, as these variables are well-established confounders in burnout research[15,17,18]The final models were developed using stepwise regression, balancing model fit and complexity. This method iteratively adds and removes variables to identify the most parsimonious model, ensuring that only variables that significantly contribute to the model are retained.

Binary logistic regression was used to calculate the odds ratios (OR) and 95% confidence intervals (CI) for each outcome. The confidence intervals were utilized to interpret the strength of the associations, providing a measure of precision and reliability. For all statistical analyses, a two-tailed P-value $< 0.05$ was deemed statistically significant.

## Results

A total of 12,343 members of the Brazilian Society of Anesthesiology (SBA) were invited to participate in the study through email invitations. These invitations were sent on three occasions: 21 June 2021, 17 January 2022, and 16 September 2022. Out of those invited, 1,351 anesthesiologists and residents completed the questionnaire, yielding a response rate of 10.94%. The average time taken to complete the questionnaire was approximately 7 minutes. Among the respondents, 1,085 (8.8% of the total invited) met the inclusion criteria (Fig 1). An infographic summarizing the key findings of the study has been included to provide a clear and visual overview of the results. This is available as S2 File.

## Participant characteristics

Our sample consisted of a similar proportion of male and female participants, accounting for 53.3% and 46.7%, respectively. Notably, the proportion of female members in the SBA during the study period was relatively lower, with 37.8% female and 62.2% male. The age distribution among the study participants closely resembled that of the overall SBA membership, with the majority falling within the 30–39 age group. Most respondents were married (59.9%), had children (56.1%), and worked in the Southeast region of the country (36.86%). Table 1 provides a comprehensive overview of the distribution of sociodemographic variables among individuals who met the criteria for burnout syndrome or were at high risk for burnout.

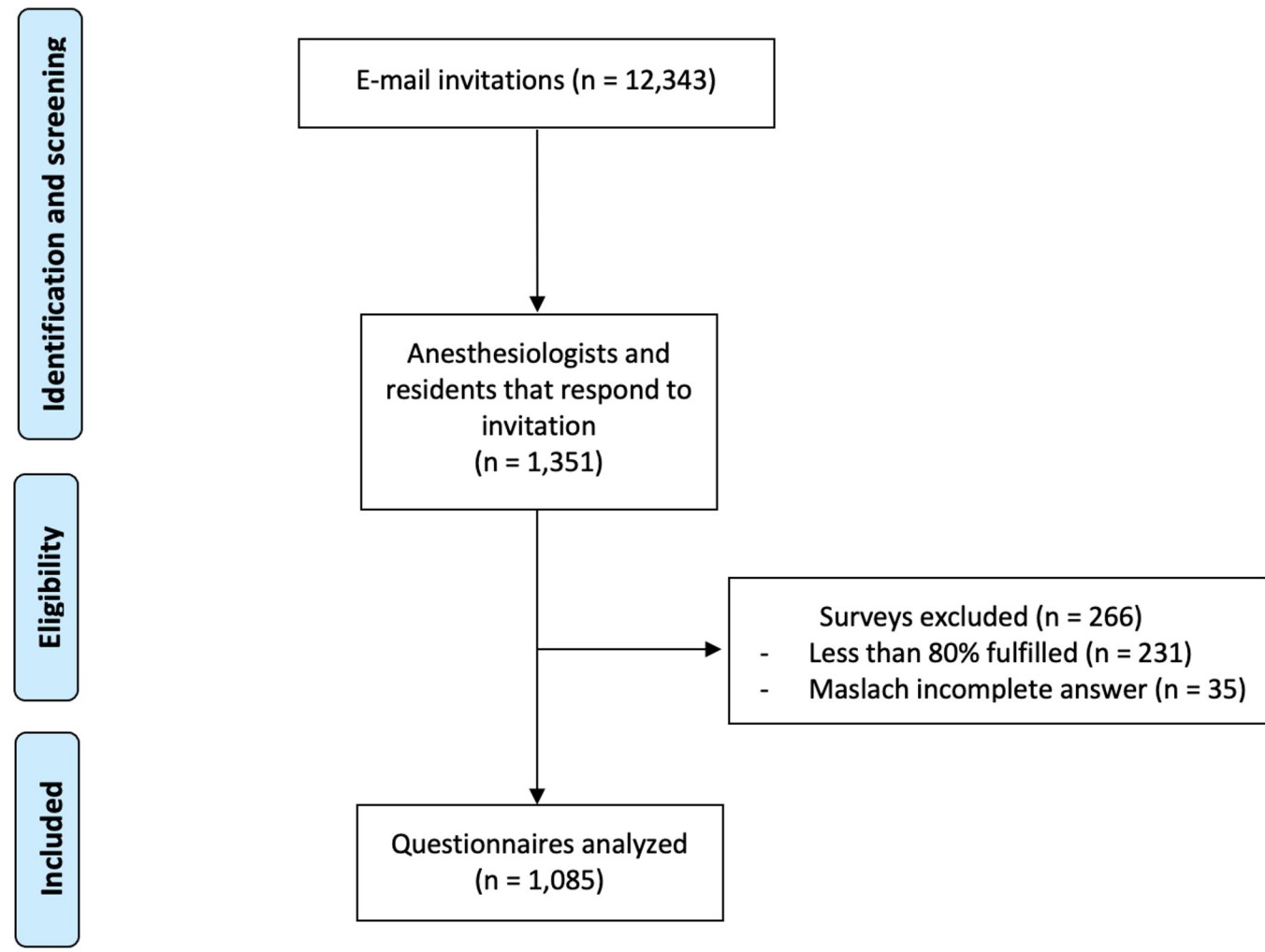

**Fig 1. Data collection flowchart.**

### Prevalence of burnout syndrome and high risk for burnout

Out of the 1,085 participants included in the study, 213 (19.63%) met the criteria for burnout syndrome. When examining the individual domains, 524 (48.2%) participants scored $\geq$ 27 in emotional exhaustion (EE), 396 (36.4%) scored $\geq$ 10 in depersonalization (DE), and 401 (36.9%) scored $\leq$ 33 in reduced personal accomplishment (RRP). Furthermore, 613 (56.49%) participants were identified as being at high risk for burnout, as they met the criteria of scoring $\geq$ 27 in EE and/or $\geq$ 10 in DE (Fig 2).

Among the participants who met the criteria for burnout (213 individuals), the majority (56.3%) reported working more than 60 hours per week. However, there was no significant difference in the number of institutions they worked in (p = 0.20). Interestingly, 43.1% of individuals with burnout scores dedicated less than five hours per week to leisure activities. A significant proportion of anesthesiologists with burnout criteria (74.6%) revealed that they had contemplated giving up their specialty.

In the logistic regression model for assessing the association with burnout, the following variables were included age, sex, have children, years of anesthetic practice, hours of work per week, night time shifts, work at weekends, hours of leisure per week, thoughts about quitting

**Table 1. Prevalence of burnout and high risk of burnout according to sociodemographic and work characteristics of study participants.** Comparisons were made between different sociodemographic and work characteristics using Pearson's χ2 test for categorical variables. Significance was determined based on a two-tailed P-value < 0.05.

| Variables | | No (N = 872) | Yes (N = 213) | p-value | No (N = 472) | Yes (N = 613) | p-value |
|---|---|---|---|---|---|---|---|
| | | **Burnout** | | | **High risk** | | |
| **Age** | < 30 yr | 103 (11.8%) | 38 (17.8%) | <0.001 | 42 (8.9%) | 99 (16.2%) | <0.001 |
| | 30–39 yr | 334 (38.3%) | 99 (46.5%) | | 153 (32.4%) | 280 (45.7%) | |
| | 40–49 yr | 174 (20%) | 41 (19.3%) | | 102 (21.6%) | 113 (18.4%) | |
| | > 50 yr | 261 (29.9%) | 35 (16.4%) | | 175 (37.1%) | 121 (19.7%) | |
| **Sex** | Male | 472 (54.1%) | 106 (49.8%) | 0.25 | 287 (60.8%) | 291 (47.5%) | <0.001 |
| | Female | 400 (45.9%) | 107 (50.2%) | | 185 (39.2%) | 322 (52.5%) | |
| **Children** | No | 352 (40.4%) | 125 (58.7%) | <0.001 | 149 (31.6%) | 328 (53.5%) | <0.001 |
| | Yes | 520 (59.6%) | 88 (41.3%) | | 323 (68.4%) | 285 (46.5%) | |
| **Number of Hospital affiliations** | 1–2 | 365 (41.8%) | 92 (43.2%) | 0.20 | 199 (42.2%) | 258 (42.1%) | 0.39 |
| | 3–4 | 339 (38.9%) | 91 (42.7%) | | 179 (37.9%) | 251 (40.9%) | |
| | + 5 | 168 (19.3%) | 30 (14.1%) | | 94 (19.9%) | 104 (17%) | |
| **Years of anesthetic practice** | < 5 | 211 (24.2%) | 88 (41.3%) | <0.001 | 77 (16.4%) | 222 (36.3%) | <0.001 |
| | 5–15 | 276 (31.7%) | 64 (30%) | | 147 (31.1%) | 193 (31.5%) | |
| | 16–25 | 158 (18.1%) | 34 (16%) | | 93 (19.7%) | 99 (16.1%) | |
| | > 25 | 227 (26%) | 27 (12.7%) | | 155 (32.8%) | 99 (16.1%) | |
| **Hours of work per week** | < 45 | 197 (22.6%) | 27 (12.7%) | <0.001 | 138 (29.3%) | 86 (14%) | <0.001 |
| | 46–60 | 294 (33.7%) | 62 (29.1%) | | 170 (36%) | 186 (30.4%) | |
| | > 60 | 364 (41.7%) | 120 (56.3%) | | 154 (32.6%) | 330 (53.8%) | |
| | NR | 17 (2%) | 4 (1.9%) | | 10 (2.1%) | 11 (1.8%) | |
| **Night time shifts** | No | 214 (24.5%) | 36 (16.9%) | 0.018 | 139 (29.5%) | 111 (18.1%) | <0.001 |
| | Yes | 658 (75.5%) | 177 (83.1%) | | 333 (70.5%) | 502 (81.9%) | |
| **Work at weekends** | No | 141 (16.2%) | 20 (9.4%) | 0.013 | 99 (21%) | 62 (10.1%) | <0.001 |
| | Yes | 728 (83.5%) | 192 (90.1%) | | 372 (78.8%) | 548 (89.4%) | |
| | NR | 3 (0.3%) | 1 (0.5%) | | 1 (0.2%) | 3 (0.5%) | |
| **Live with** | Friends/ Family | 742 (85.1%) | 178 (83.6%) | 0.58 | 421 (89.2%) | 499 (81.4%) | <0.001 |
| | Alone | 130 (14.9%) | 35 (16.4%) | | 51 (10.8%) | 114 (18.6%) | |
| **Hours of leisure per week** | ≤ 5 | 210 (24.1%) | 92 (43.2%) | <0.001 | 81 (17.2%) | 221 (36.0%) | <0.001 |
| | 6 to 10 | 359 (41.2%) | 89 (41.8%) | | 184 (38.9%) | 264 (43.1%) | |
| | 11 to 20 | 164 (18.8%) | 22 (10.3%) | | 106 (22.5%) | 80 (13.1%) | |
| | > 21 | 139 (15.9%) | 10 (4.7%) | | 101 (21.4%) | 48 (7.8%) | |
| **Thoughts about quitting the specialty** | No | 545 (62.5%) | 54 (25.4%) | <0.001 | 359 (76.1%) | 240 (39.2%) | <0.001 |
| | Yes | 327 (37.5%) | 159 (74.6%) | | 113 (23.9%) | 373 (60.8%) | |
| **Region** | Southeast | 395 (45.3%) | 109 (51.2%) | 0.004 | 197 (41.7%) | 307 (50.1%) | <0.001 |
| | Northeast | 194 (22.2%) | 28 (13.1%) | | 125 (26.5%) | 97 (15.8%) | |
| | North | 34 (3.9%) | 9 (4.2%) | | 14 (2.9%) | 29 (4.7%) | |
| | Center-west | 81 (9.3%) | 30 (14.1%) | | 44 (9.4%) | 67 (11%) | |
| | South | 168 (19.3%) | 37 (17.4%) | | 92 (19.5%) | 113(18.4%) | |
| **Marital status** | Married | 524 (60.1%) | 106 (49.7%) | <0.002 | 310 (65.7%) | 320 (52.2%) | <0.001 |
| | Single | 187 (21.4%) | 70 (32.9%) | | 72 (15.2%) | 185 (30.2%) | |
| | Other | 161 (18.5%) | 37 (17.4%) | | 90 (19.1%) | 108 (17.6%) | |

the specialty, geographic region, marital status All variables demonstrated a statistically significant association with burnout in the univariable analysis, except for sex (p-value = 0.25).

When examining the high risk for burnout, the prevalence rates were notably higher among female participants (52.5%), those without children (53.5%), individuals residing in the

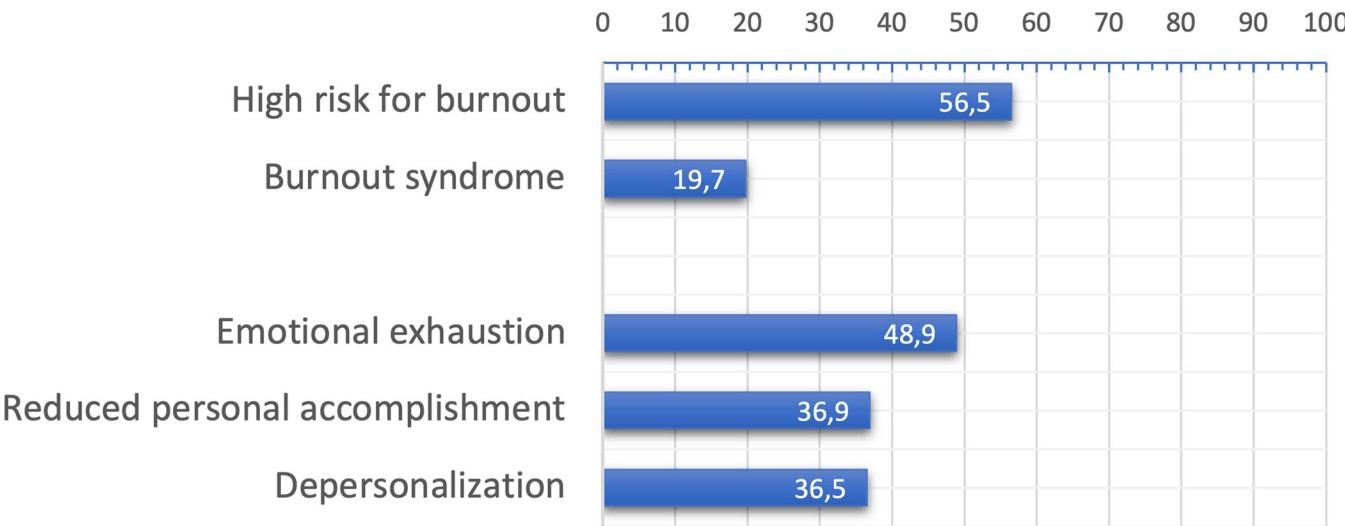

**Fig 2. Prevalence of burnout syndrome and high risk for burnout.**

Southeast region of the country (50.1%), those with less than five years of anesthesia practice (36.2%), individuals spending six to ten hours on leisure activities (43.1%), and those who had considered quitting the specialty (60.8%). All these variables exhibited a statistically significant association with the high risk for burnout (p-value < 0.05).

Multivariable logistic regression, adjusted for other variables in the model, are presented in Fig 3. Among the 12 variables included in the logistic regression, only five showed statistical significant association with burnout syndrome: anesthesiologists with children had a 41% lower chance of experiencing burnout (OR: 0.59 (95% CI 0.36–0.96)); leisure time of more than 21 hours per week showed a 74% lower chance (OR: 0.26 (95% CI 0.12–0.52)) compared to less than five hours, with a directly proportional relationship to the number of leisure hours; anesthetic practice duration between 16 and 25 years presented a 60% lower chance (OR: 0.40 (95% CI 0.16–0.99)); working in the Northeast region of Brazil reduced the chance by 39% (OR: 0.61 (95% CI 0.37–0.99)), and having considered giving up the specialty was the most strongly related factor (OR: 4.72 (95% CI 3.30–6.83)). The multivariable logistic regression model explained approximately 24% of the variance in burnout (Nagelkerke's $R^2$ = 0.24), indicating that the included factors accounted for a significant portion of the variability in burnout among participants. A table presenting the multivariable analysis of factors associated with Burnout Syndrome and the high-risk development is attached (S3 File).

When analyzing the association with high risk for burnout, six risk factors presented statistical significance, four of which were also associated with burnout syndrome: longer duration of anesthetic practice was directly and inverse proportionally related, with more than 25 years reducing the chance by 70% (OR: 0.30 (95% CI 0.10–0.85)); longer leisure time was also directly and proportionally related, with more than 21 hours per week reducing the risk by 69% (OR: 0.31 (95% CI 0.19–0.50)); working in the Northeast region reduced the chance by 51% (OR: 0.49 (95% CI 0.33–0.71)), and having considered giving up the specialty was also the most strongly implicated variable (OR: 4.99 (95% CI 3.70–6.77)). In addition, the other two factors related to high risk for burnout were being female (OR: 1.74 (95% CI1.28–2.36)) and working more than 60 hours per week (OR: 2.35 (95% CI 1.52–3.65)). The multivariable logistic regression model for high risk for burnout explained approximately 35% of the variance in high risk for burnout (Nagelkerke's $R^2$ = 0.35).

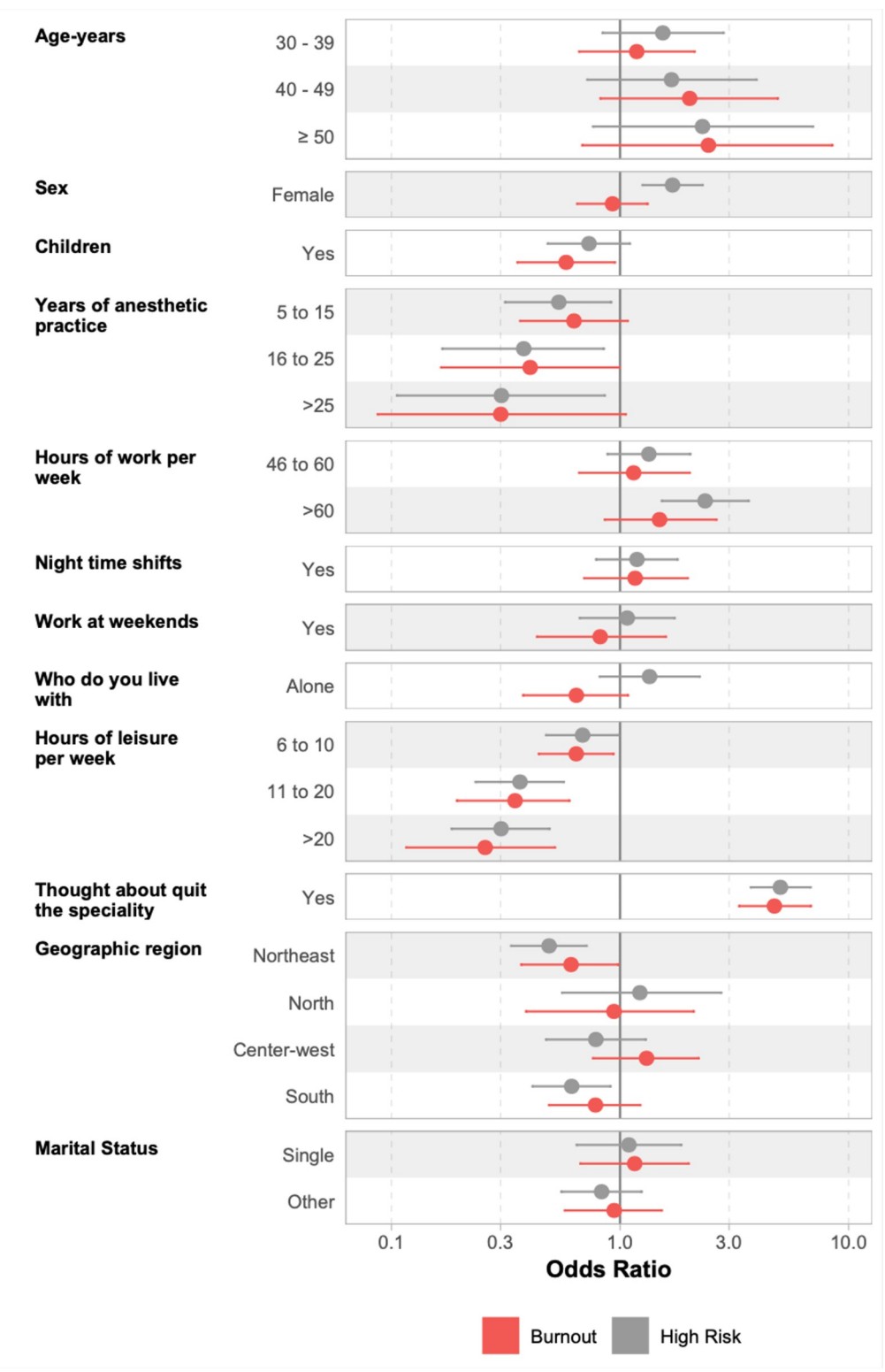

**Fig 3. Multivariable logistic regression of factors associated with burnout syndrome and high-risk development.**
References category: age = < 30 ys; sex = male; children = no; years of anesthetic practice = < 5; hours of work peer week = ≤ 45; nighttime shifts = no; work at weekends = no; live with = friends/family; hours of leisure pr week = ≤ 5; thoughts about quitting the speciality = no; geographic region = Southeast; marital status = married.

## Discussion

### Main findings

This is the first study to examine burnout among all Brazilian anesthesiologists and anesthesiology residents during the COVID-19 pandemic, providing a representative proportional sample based on data from the Brazilian Society of Anesthesiology (SBA). Our findings reveal a concerning reality: over half of the respondents (56.49%) were at high risk for burnout, and 19.63% met the criteria for burnout syndrome. These results highlight the substantial burden faced by anesthesiologist during this critical period.

### Global burnout scenario

Our findings align with international data, reinforcing the global nature of burnout crisis among anesthesiologists. A nationwide study involving 3,898 American Society of Anesthesiologists (ASA) members reported very similar burnout patterns, with 59,2% at high risk and 13.8% experiencing full burnout syndrome [15]. The levels of emotional exhaustion (EE) and depersonalization (DE) among ASA and SBA members were also comparable, with rates of 53.3% and 37.2% (ASA) and 48.29% and 36.49% (SBA), respectively, suggesting shared environmental factors.

In contrast, burnout rates among anesthesia providers in Zambia reached 51.3% before the COVID-pandemic, although this figure includes non-physician providers, which may account for the higher prevalence [16]. However, including such comparisons can offer valuable insights into the diverse factors that contribute to burnout across different healthcare environments. Burnout is a multifaceted issue influenced by various factors, including workload, resources, healthcare infrastructure, and cultural attitudes towards work and stress. By comparing our findings with those from other countries, we aim to highlight the global nature of the burnout problem and emphasize that it is not confined to a single region or set of circumstances. Furthermore, the prevalence of burnout among Chinese anesthesiologists was even higher, reaching 52.7% in a recent large-scale study during COVID-pandemic [17].

### Variables associated with burnout and higher burnout risk

Several key variables were identified as significant predictors of burnout. In this study, although high risk and burnout were more prevalent among younger individuals compared to those above 50 years old, age itself was not found to be a significant factor for burnout or higher risk. While some studies have implicated age as a contributing factor, [15] the focus of research has primarily been on the period of anesthesia practice rather than age per se. Early-career anesthesiologists face unique challenges during their transition period, characterized by a combination of limited experience in anesthesia practice and increased responsibilities upon completing their residency. Factors such as adapting to new hospital environments, navigating hospital protocols, financial concerns, and personal stressors can all contribute to developing work-life challenges and conflicts [18].

Residency, as a phase of professional development, has not been directly associated with a higher risk of burnout compared to experienced anesthesiologists, regardless of whether it is assessed before or after the pandemic [19–22]. A meta-analysis conducted prior to the pandemic concluded that burnout was not a widespread problem among resident physicians [23]. Even so, younger participants had a higher risk of experiencing burnout syndrome than those over 50. These findings align with research conducted in the United States [20] and Canada, where younger emergency physicians, specifically those aged 30–39, were at a higher risk of burnout compared to their counterparts aged 40–49 [9].

Sex has been identified as a significant factor in developing burnout symptoms, with women being more likely to experience burnout [9, 24, 25]. Additionally, studies have suggested that non-binary genders may also have a higher prevalence of burnout [15]. In the present study, nearly half of the participants were female, and they exhibited a higher prevalence of higher risk for burnout compared to their male counterparts.

The Northeast region of Brazil stands out for its cultural diversity, vibrant history, and varied climate, from tropical coastal areas to semi-arid zones. Economically, it faces challenges such as higher poverty rates and lower average incomes compared to the Southeast and South regions [26]. In healthcare, the Northeast often struggles with disparities in access and resources, with some hospitals lacking infrastructure and advanced equipment [27]. Despite these challenges, the cost of living is generally lower, and the region's warmer climate with more sunshine year-round contributes to better overall well-being and a higher quality of life [28, 29]. These factors may explain the lower prevalence of burnout among anesthesiologists in the Northeast, suggesting a unique balance of work-life conditions. Further research is needed to explore these associations in more detail.

Despite the comprehensive data presented, it is important to note that there were no available data on the burnout scenario in Brazil prior to the pandemic, which limits the ability to make direct comparisons with the pre-COVID-19 period.

## Impact of COVID-19 on the work environment and burnout

The COVID-19 pandemic placed unprecedented strain on healthcare systems, particularly for anesthesiologists, who were often on the front lines treating critically ill patients [30]. While some studies did not observed a significant increase in burnout rates during pandemic, the full impact of pandemic has yet to be understood [19]. Similarly, a recent review assessing the impact of COVID-19 on physicians also failed to reveal significant differences [7].

In our study, 59% of participants reported that the pandemic did not influence their responses, which may suggest resilience or the delayed manisftation of stress-related symptoms. Nevertheless, it is important to note that it might still be too early to comprehensively assess the full impact. Since anesthesiologists possess diverse skills applicable to various healthcare settings beyond the operating room (e.g., ICU, emergency departments, and complex patient cases), the consequences of the additional workload may only become apparent with more time elapsed. Interestingly, a recent cross-sectional survey involving members of the Society of Critical Care Anesthesiologists found no significant association between COVID-19 workload and burnout, suggesting that the pandemic's impact on professional self-realization may be limited [31].

When comparing our findings with pre-pandemic data available in Brazil, [12, 13] the prevalence rates significantly increased, suggesting that the COVID-19 pandemic may have exacerbated this issue. Additionally, when comparing to the results of data available during pandemic, [32] which used a the Oldenburg Burnout Inventory, their findings indicated a 73.2% risk of burnout among anesthesiology residents, with 57.1% at high risk. While their study focused on residents and used a different instrument, the elevated burnout rates in both studies suggest that the pandemic intensified burnout risk.

## Burnout consequences

In the scenario of a crushing workload for physicians, a recent survey evaluated the exodus of this career during the COVID-19 pandemic, with 23.8% of over 9,000 physicians from various specialties planning to leave primary care within three years [33]. In line with these findings, the present study revealed that physicians that have consider leaving the profession in the

coming years are five times more likely to have burnout and this factor emerges as its primary contributor. While individual factors play a role, the nature of the specialty itself, characterized by long working hours and the stress associated with managing critically ill patients, may contribute significantly to the higher incidence of burnout. Moreover, burnout can lead to various behavioral and psychological changes, including attention deficit disorders, depression, increased risk of other mental disorders, substance abuse, poor patient outcomes, and suicide [34–37].

### Mitigation strategies: Promoting the virtuous cycle to prevent burnout

Adressing the high prevalence of burnout requires a multifaced approach. Leadership withing healthcare institution plays a crutial role in recognizing and addressing burnout symptoms and fostering well-being in the workplace can significantly increased job satisfaction [38].

One of the most extensively studied practices for stress management is mindfulness [39]. A systematic review confirmed the positive impact of mindfulness of mindfulness on empathy, well-being, and reduction of burnout in physicians [40].

At the institutional level, hospitals need to be attentive to professionals' physical well-being, promote healthy eating, provide adequate breaks and rest areas, and limit excessive workload [8]. Psychological support, such as cognitive-behavioral therapy, can be a valuable long-term strategy, as single sessions have not shown significant benefits [41]. Offering free and frequent psychiatric and psychological assessments for physicians is suggested as a strategy for early diagnosis of mental health issues [42].

### The Brazilian experience

In early 2022, with the implementation of ICD-11 globally, burnout was officially recognized by the World Health Organization (WHO) as an occupational disease. In April 2020, the SBA initiated a self-care and mental health promotion program called "Núcleo do Eu" (Core of Self). Through a series of virtual and free strategies offered to its members, activities such as practicing gratitude, forgiveness, spirituality, mindfulness, physical exercise, healthy nutrition, and good financial management are encouraged to foster the virtuous cycle. These initiatives are carried out both virtually, through monthly webinars since 2020, and in-person, as the "Núcleo" provides free workshops, group activities, and individual sessions at all official SBA events, including a focus on financial well-being. Although tangible data on the impact of these strategies on the mental health of SBA members is not yet available, it is believed that objective results will be achieved soon.

### Recommendations for practice and future research

To mitigate burnout among anesthesiologists, we recommend the following:

1. **Implement Comprehensive Mental Health Programs:** Expand initiatives like the SBA's "Núcleo do Eu", which promotes mental health and well-being [42].

2. **Promote Mindfulness and Stress Management Techniques:** Integrate mindfulness training into the routine of anesthesiology professionals [39, 40].

3. **Enhance the Work Environment:** Ensure adequate breaks, reasonable work hours, and access to healthy food in healthcare settings [40, 42].

4. **Provide Continuous Psychological Support:** Offer accessible psychiatric and psychological services for ongoing mental health support [43].

5. **Conduct Longitudinal Studies:** Further research is needed to monitor the long-term effects of burnout and the effectiveness of interventions.

## Study limitations

Our study has several limitations. Firstly, the cross-sectional survey was conducted online on a voluntary basis, which may introduce selection bias as it was restricted to SBA members with updated data, and the topic of burnout may attract respondents with a higher interest in the subject. Another limitation is the low response rate of the questionnaires in our sample, representing only 10.9% of all SBA members. Additionally, it was a limitation of this study that the questionnaire did not include an option for non-binary gender, which has been previously associated with a higher risk of the syndrome.[13] The reliance on self-reported outcomes and high rates of dropouts or non-participation have been major limitations in studies involving burnout, especially in the context of COVID.[38] Moreover, we did not perform a comparison of burnout rates between the three groups based on the timing of their responses to the invitations. This type of subgroup analysis could provide valuable insights into temporal variations in burnout levels, and we recommend this as a potential area for future research.

## Conclusion

This study represents a comprehensive assessment of burnout among all Brazilian anesthesiologists during the COVID-19 pandemic, highlighting a high prevalence of burnout and its associate risk factors. When compared to pre-pandemic data from Brazil, our findings indicate a notable increase in burnout rates among anesthesiologists during the COVID-19 pandemic, highlighting the exacerbating effect of the pandemic on mental health in this professional group. This underscores the urgent need for targeted interventions to mitigate burnout and promote mental health and well-being among anesthesiologists. The findings provide a foundation for the Brazilian Society of Anesthesiology to enhance its mental health policies and offer valuable insights for healthcare managers, residency program directors, and anesthesiologists. Future research should focus on longitudinal studies to monitor the effectiveness of these interventions and explore the long-term impact of burnout in this population.

## Supporting information

**S1 File. Socio-demographic and psychosocial questionnaire used during data collection.**
(PDF)

**S2 File. Infographic summarizing the key findings of the study.**
(PDF)

**S3 File. Table presenting the multivariable analysis of factors associated with burnout syndrome and high-risk development.**
(PDF)

**S1 Data.**
(XLSX)

## Author Contributions

**Conceptualization:** Liana M. T. A. Azi, Thaiane S. Ferreira.

**Data curation:** Thiago Cerqueira-Silva, Matheus L. Azi.

**Formal analysis:** Thiago Cerqueira-Silva, Matheus L. Azi.

**Investigation:** Liana M. T. A. Azi, Thaiane S. Ferreira, Luis A. S. Diego, Marcos A. C. Albuquerque.

**Methodology:** Liana M. T. A. Azi, Thaiane S. Ferreira, Thiago Cerqueira-Silva, Matheus L. Azi.

**Project administration:** Liana M. T. A. Azi, Matheus L. Azi.

**Resources:** Luis A. S. Diego, Marcos A. C. Albuquerque.

**Supervision:** Liana M. T. A. Azi, Marcos A. C. Albuquerque, Matheus L. Azi.

**Writing – original draft:** Liana M. T. A. Azi, Thaiane S. Ferreira.

**Writing – review & editing:** Liana M. T. A. Azi, Thiago Cerqueira-Silva, Luis A. S. Diego.

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
