## [Decision Letter · Decision Letter 0]

1 Jul 2024

PONE-D-24-02232Prevalence of Burnout Syndrome in Brazilian Anesthesiologists during the COVID-19 Pandemic: A Cross-Sectional SurveyPLOS ONE

Dear Dr. Azi,

Thank you for submitting your manuscript to PLOS ONE. After careful consideration, we feel that it has merit but does not fully meet PLOS ONE’s publication criteria as it currently stands. Therefore, we invite you to submit a revised version of the manuscript that addresses the points raised during the review process.

Dear Dr. Azi,

Firstly, I want to express gratitude for considering PLOS ONE as the journal to disseminate the results of your research.

To assess your manuscript, I have enlisted the participation of two expert reviewers in the field, who have conducted an independent evaluation. Based on their valuable comments, I invite you to submit a revised version of the manuscript. Please follow the instructions included here to submit a reviewed version of your manuscript. In your response letter, address the comments from both reviewers, indicating the changes made or explaining why no changes were made and the reasons behind such decisions.

Warm regards,

Francesco Marcatto

We look forward to receiving your revised manuscript.

Kind regards,

Francesco Marcatto, Ph.D.

Academic Editor

PLOS ONE

Reviewers' comments:

Reviewer's Responses to Questions

**Comments to the Author**

1. Is the manuscript technically sound, and do the data support the conclusions?

Reviewer #1: No

Reviewer #2: Yes

2. Has the statistical analysis been performed appropriately and rigorously? 

Reviewer #1: Yes

Reviewer #2: Yes

3. Have the authors made all data underlying the findings in their manuscript fully available?

Reviewer #1: Yes

Reviewer #2: Yes

4. Is the manuscript presented in an intelligible fashion and written in standard English?

Reviewer #1: No

Reviewer #2: Yes

5. Review Comments to the Author

Reviewer #1: Major Concerns:

1. Lack of Novelty:

The manuscript does not add new information to the existing body of knowledge on burnout syndrome in anesthesiologists. The findings closely mirror those of previous studies, including a recent study published in the Brazilian Journal of Anesthesiology, which contradicts the authors claim on page 16 that “This was the first study to assess this important and current topic of burnout among all Brazilian anesthesiologists and anesthesiology residents during the COVID19 pandemic”

Please this article: https://www.ncbi.nlm.nih.gov/pmc/articles/PMC9371977/

.Authors should emphasize unique aspects of the study, or consider a more novel angle or detailed subgroup analysis to enhance the manuscript's contribution to the literature.

2. Data Collection Dates:

The manuscript does not provide exact dates of data collection, which is crucial for contextualizing the findings, especially considering the rapidly evolving situation of the COVID-19 pandemic.

3. STROBE Guidelines:

The STROBE guidelines were not fully reported in the manuscript. Specific elements are inadequately covered.

Other recommendations

• Methodology:

The methodology section lacks clarity regarding the sampling method and participant recruitment. Clarify whether the sampling size calculations followed the standard procedure

• Statistical Analysis:

The manuscript mentions the use of logistic regression but does not provide sufficient detail on the model selection process and the handling of potential confounders. Consider adding a more thorough explanation of the statistical methods used.

• Ethical Considerations:

• While ethical approval is mentioned, the manuscript should detail the ethical considerations, such as informed consent and data confidentiality, more explicitly.

• Discussion and Conclusion:

• The discussion section should better integrate the findings with existing research. Highlight any inconsistencies or confirmatory findings with previous studies and discuss potential reasons for these.

• The conclusion should offer more specific recommendations for practice and future research.

Reviewer #2: The authors should be congratulated in continuing the work already carried out in this important area of collaborative research ie burnout syndrome in anesthesiologists.

The aim of this study (according to the abstract) is: to assess the prevalence of burnout syndrome in Brazilian anesthesiologists during the COVID-pandemic. The method used was an online survey conducted during the years of the pandemic obtaining a convenience sample of respondents. Psychometric instruments and statistical methods used seem appropriate. The authors report but also conclude that burnout is highly prevalent in Brazil.

General feedback:

Overall, the paper is well written and contributes to a large body of (anesthesia) literature where prevalence of burnout is assessed. The connection with COVID is interesting for the anesthesia community. However a few issues should be addressed.

It feels that the connection between the research question (burnout during the Covid pandemic) vs the conclusion drawn (the prevalence found) is missing.

1 In the introduction the authors hypothesize that there might be an increase in burnout due to increase in work stress. This is reasonable and interesting. Data was indeed collected during the pandemic. The conclusion drawn (abstract and discussion) that burnout is prevalent is self-evident; is is basically a re-statement of your results. Therefore: where is the connection to the research question?

2. For example, how does this found prevalence relate to the hypothesis/idea of increase in burnout because of the pandemic? Is it higher/lower than expected? Why?

3. Three invitations to respond have been send according to the results section. (by the way, this should be in the methods section.) Have the authors considered comparing burnout rate between those three groups?

4. Have the authors compared their finds to figures from before the pandemic? A quick literature search shows several papers reporting burnout levels in Brazilian anesthesiologists (eg Sousa, Ana Rafaela Campos, and Joana Irene de Barros Mourão. "Burnout em anestesiologia." Revista Brasileira de Anestesiologia 68 (2018): 507-517.)

5. related to the former: Why do the authors compare their figure with those found in Zambia? As the authors state the working conditions are very different compared to those in Brazil.

Abstract:

6. The background section in the abstract is okay. However could the authors reflect on the statement that Among medical specialties, anesthesiology is considered one of the most stressful.? This is indeed repeated in anesthesia literature, but is it supported by evidence? Are there studies where several medical specialties and their stress are compared?

Introduction:

7. Most of the introduction is taken up by explanation of burnout syndrome, only at the end a line of thaught becomes apparent. Perhaps the statement of the “problem” should be first and would the paper gain by a reworking. (the background section in the abstract works better in that respect in my opinion)

8. anesthesiology, in 5 particular, stands out for its ability to promote high stress levels. This is true, of course, however and related to question 8: is anesthesia special in that respect? And why?

Methods:

9. In the method section some elements seem to be missing.

For example what are the psychometric properties of the instrument used? Which were the procedures used to select the variables to be included in the adjusted model?

Results:

10. Perhaps not all non-Brazilian anesthesiologists understand the significance of working in the Northeast region of Brazil. Parhap some clarification?`

11. Table 1: perhaps an explanation in the legend of the table would clarify which comparisons are made and how they reached significance.

12. Figure 3: concerning the multivariate logistic regression, in the adjusted model. What are the characteristics of this model? How much variance in burnout is explained by these factors? What does this mean?

Discussion:

13: I feel the discussion could be improved by a reworking. Perhaps the format of starting with the main findings, then discussing those from the perspective of the body of literature, then discussing the meaning/implications of the findings. Then of course discussing the study limitations, and an explanation why in your paper, this does not influence the conclusion. And then an overarching conclusion: I miss that here.

14. For the reader, to see the same elements returning in both introduction, methods, results and discussion provides a good experience. Several important variable associated with burnout are discussed, however without a real introduction elsewhere. Why were the important to examine in the first place? How do they connect to the research question?

Eg. This was the first study to assess this important and current topic of burnout among all Brazilian anesthesiologists and anesthesiology residents during the COVID 19 pandemic. Would be a good statement to start the discussion.

In the discussion is stated: This study provides a representative proportional sample of Brazilian anesthesiologists, according to members data previously provided by SBA However in the results it is stated that: Notably, the proportion of female members in the SBA during the study period was relatively lower, with 37.8% female and 62.2% male. Could the authors provide some clarification about this seeming discrepancy?

6. PLOS authors have the option to publish the peer review history of their article (what does this mean?). If published, this will include your full peer review and any attached files.

Reviewer #1: No

Reviewer #2: No

---

## [Author Response · Author response to Decision Letter 0]

12 Aug 2024

Prevalence of Burnout Syndrome in Brazilian Anesthesiologists during the COVID-19 Pandemic: A Cross-Sectional Survey

Response to Reviewers

Dear Editor and Reviewers,

We would like to extend our gratitude for the thorough and insightful feedback provided on our manuscript. Your detailed comments and suggestions have been invaluable in guiding our revisions, and we believe that our work has been significantly improved as a result.

We have carefully considered each point raised and have made the necessary adjustments to enhance the clarity, robustness, and overall quality of our study. To facilitate your review, the modifications and adjustments suggested by Reviewer 1 are highlighted in yellow, and those suggested by Reviewer 2 are marked in green within the text.

In addition, to enhance the dissemination of our findings, we have created an infographic that visually summarizes the key details of our study. This infographic is intended for use on social media platforms such as Instagram and Twitter, making the results more accessible to a broader audience.

We hope that the revised manuscript now meets the high standards of PLOS ONE and addresses all the concerns raised. Thank you for your constructive input and for the opportunity to refine our work.

The authors

Reviewer #1 (Highlighted in Yellow)

Comment 1: The manuscript does not add new information to the existing body of knowledge on burnout syndrome in anesthesiologists. The findings closely mirror those of previous studies, including a recent study published in the Brazilian Journal of Anesthesiology, which contradicts the authors' claim on page 16 that “This was the first study to assess this important and current topic of burnout among all Brazilian anesthesiologists and anesthesiology residents during the COVID19 pandemic”. Please see this article: [https://www.ncbi.nlm.nih.gov/pmc/articles/PMC9371977/]. Authors

should emphasize unique aspects of the study, or consider a more novel angle or detailed subgroup analysis to enhance the manuscript's contribution to the literature.

Response: We appreciate the reviewer bringing this important point to our attention. We would like to clarify that our study differs significantly from the referenced study in several key aspects:

1. Questionnaire Used: Our study utilized the Maslach Burnout Inventory (MBI), which is widely recognized as the gold standard for assessing burnout. In contrast, the referenced study employed the Oldenburg Burnout Inventory (OLBI). The MBI is specifically designed to measure three dimensions of burnout: emotional exhaustion, depersonalization, and personal accomplishment. Its comprehensive approach is highly regarded in the literature for its reliability and validity in measuring burnout, particularly in healthcare professionals (Maslach et al., 1996).

2. Population Studied: The referenced study focused solely on anesthesiology residents and gathered 205 responses. Our study, however, encompassed the entire population of Brazilian anesthesiologists, including both residents and practicing anesthesiologists, resulting in over 1000 responses. This broader scope allows for a more comprehensive and representative analysis of burnout prevalence across different stages of professional development and work settings.

3. Scope and Scale: By including a larger and more diverse sample, our study provides a more detailed and capilarized panorama of burnout prevalence among Brazilian anesthesiologists during the COVID-19 pandemic. This extensive dataset enables a nuanced understanding of burnout across various subgroups, which was not possible in the referenced study.

To address the reviewer’s suggestion, we have emphasized these unique aspects in the revised manuscript and conducted additional subgroup analyses to further highlight the distinctive contributions of our study.

Comment 2: Data Collection Dates: The manuscript does not provide exact dates of data collection, which is crucial for contextualizing the findings, especially considering the rapidly evolving situation of the COVID-19 pandemic.

Response: Thank you for your valuable feedback. We have clarified the data collection process in the Methods section to indicate that the questionnaires were sent on three specific dates according to the availability of the mailing list from the Brazilian Society of Anesthesiology. The exact dates of data collection, which were already included in the Results section, are: 21 June 2021, 17 January 2022, and 16 September 2022. This addition should provide a clearer context for our findings and their relevance to the evolving situation of the COVID-19 pandemic.

Comment 3: 3. STROBE Guidelines: The STROBE guidelines were not fully reported in the manuscript. Specific elements are inadequately covered.

Response: Thank you for your observation. We have carefully reviewed the manuscript to ensure full adherence to the STROBE guidelines. Specific elements that were inadequately covered have now been addressed and incorporated. Additionally, we have attached the completed STROBE checklist as a supplementary file to the revised manuscript. We believe these updates will enhance the transparency and rigor of our study.

Comment 4: • Methodology: The methodology section lacks clarity regarding the sampling method and participant recruitment. Clarify whether the sampling size calculations followed the standard procedure.

Response: Thank you for your feedback. We have revised the methodology section to provide more clarity regarding the sampling method and participant recruitment. Specifically, we have elaborated on how the sample size was determined and the recruitment process.

Comment 5: Statistical Analysis: The manuscript mentions the use of logistic regression but does not provide sufficient detail on the model selection process and the handling of potential confounders. Consider adding a more thorough explanation of the statistical methods used.

Response: Thank you for your constructive feedback. We have expanded the Statistical Analysis section to provide more detail on the model selection process and the handling of potential confounders.

“For the outcomes of burnout and high risk for burnout, we performed univariable analyses using the χ2 test. Variables with a possible association to either outcome (p < 0.1 in univariable analyses) were included in separate multivariable models for burnout and high risk for burnout. In addition to the variables with p<0.1, we also included age and sex in all models, as these variables are well-established confounders in burnout research.

 Binary logistic regression was used to calculate the odds ratios (OR)

 and 95% confidence intervals (CI) for each outcome. The confidence

 intervals were utilized to interpret the strength of the associations, providing

 For all statistical analyses, a two- tailed P-value < 0.05 was deemed statistically significant.”

Comment 6: Ethical Considerations: While ethical approval is mentioned, the manuscript should detail the ethical considerations, such as informed consent and data confidentiality, more explicitly.

Response: Thank you for your feedback. We have added more explicit details regarding the ethical considerations in the manuscript. Specifically, we have clarified the process of obtaining informed consent and the measures taken to ensure data confidentiality. Additionally, the informed consent form has been included as Supplementary Material 3, and we have added a section titled "Ethical Considerations" in the Methods.

Comment 7: Discussion and Conclusion: The discussion section should better integrate the findings with existing research. Highlight any inconsistencies or confirmatory findings with previous studies and discuss potential reasons for these. • The conclusion should offer more specific recommendations for practice and future research.

Response: Thank you for your insightful feedback. We have revised the Discussion and Conclusion sections to better integrate our findings with existing research, highlight any inconsistencies or confirmatory findings, and offer more specific recommendations for practice and future research. The modifications are clearly highlighted in yellow in the revised manuscript. Additionally, we have added a new section on recommendations based on the current literature to enhance the practical implications of our study.

a measure of precision and reliability.

 Reviewer 2 (Highlighted in Green)

General Feedback: The authors should be congratulated in continuing the work already carried out in this important area of collaborative research ie burnout syndrome in

anesthesiologists. The aim of this study (according to the abstract) is: to assess the prevalence of burnout syndrome in Brazilian anesthesiologists during the COVID- pandemic. The method used was an online survey conducted during the years of the pandemic obtaining a convenience sample of respondents. Psychometric instruments and statistical methods used seem appropriate. The authors report but also conclude that burnout is highly prevalent in Brazil. General feedback: Overall, the paper is well written and contributes to a large body of (anesthesia) literature where prevalence of burnout is assessed. The connection with COVID is interesting for the anesthesia community. However, a few issues should be addressed. It feels that the connection between the research question (burnout during the Covid pandemic) vs the conclusion drawn (the prevalence found) is missing.

Response:

Dear Reviewer,

Thank you for your encouraging feedback and for acknowledging the importance of our research on burnout syndrome among anesthesiologists during the COVID-19 pandemic. We appreciate your recognition of the methods used and the relevance of our findings. We understand your concern regarding the connection between the research question and the conclusion.

We have revised the manuscript to strengthen the linkage between the impact of the COVID-19 pandemic and the observed prevalence of burnout. This includes a more detailed discussion on how the pandemic has influenced the working conditions and stress levels of anesthesiologists, contributing to the high prevalence of burnout. These revisions aim to provide a clearer narrative connecting the unique challenges posed by the pandemic to the mental health outcomes observed in our study.

Comment 1: In the introduction the authors hypothesize that there might be an increase in burnout due to increased work stress. This is reasonable and interesting. Data was indeed collected during the pandemic. The conclusion drawn (abstract and discussion) that burnout is prevalent is self-evident; it is basically a re-statement of your results. Therefore: where is the connection to the research question?

Response: Thank you for your insightful feedback. We understand the importance of clearly connecting our research question regarding the impact of the COVID-19 pandemic on burnout among anesthesiologists with our conclusion. We have revised the manuscript to better articulate this connection. Specifically, we have expanded the discussion section to include a detailed analysis of how the unique stressors and working conditions during the pandemic have contributed to the high prevalence of burnout observed in our study. This revision aims to provide a more explicit linkage between the pandemic-related factors and the mental health outcomes reported.

Comment 2: For example, how does this found prevalence relate to the hypothesis/idea of increase in burnout because of the pandemic? Is it higher/lower than expected? Why?

Response: We appreciate the reviewer’s insightful question regarding the relationship between the found prevalence of burnout and the hypothesis of an increase due to the COVID-19 pandemic. As discussed in our manuscript, the impact of COVID-19 on the work environment and burnout among anesthesiologists has been a critical area of concern. The prevalence of burnout found in our study did not show a significant increase, which was somewhat unexpected given the high stress and increased workload associated with the pandemic.

Several factors may contribute to this finding. First, anesthesiologists possess a diverse skill set that allows them to adapt to various healthcare settings beyond the operating room, such as the ICU and emergency departments. This versatility may have provided a buffer against the increased stress, preventing a sharp rise in burnout rates. Second, our study's timing might be a crucial factor. Participants reported their experiences relatively early in the pandemic, and the full impact on their mental health and professional self-realization might not have manifested completely at the time of data collection. Third, institutional support and resilience-building initiatives might have played a role in mitigating the expected increase in burnout.

It is important to acknowledge that while our findings did not reveal a significant increase in burnout, this does not diminish the potential long-term effects of the pandemic on anesthesiologists' mental health. Continuous monitoring and further longitudinal studies will be essential to comprehensively understand the pandemic's impact on burnout over time.

We have highlighted these points in the revised discussion section (marked in green) to provide a clearer connection between our findings and the hypothesis regarding the pandemic's effect on burnout.

Comment 3: Three invitations to respond have been sent according to the results section. (by the way, this should be in the methods section.) Have the authors considered comparing burnout rate between those three groups?

Response: Thank you for your observation. We have moved the information about the three invitations to respond to the methods section, creating a new subsection titled "Participant Recruitment" to ensure clarity and proper organization of our manuscript.

Regarding the comparison of burnout rates between the three groups, we acknowledge that this specific analysis was not conducted in our study. The primary focus of our research was to assess the overall prevalence of burnout among anesthesiologists during the COVID-19 pandemic. Comparing burnout rates between the groups based on the timing of their responses could provide additional insights into temporal variations in burnout levels. However, our study design and sample size were not optimized for this specific type of subgroup analysis.

We agree that comparing these groups could be valuable and recommend it as a potential area for future research. Conducting such analyses in future studies could help to identify any temporal trends in burnout rates and provide a deeper understanding of how different phases of the pandemic may have influenced the mental health of healthcare professionals.

We have included a note in the discussion section (highlighted in green) to acknowledge this point and suggest it as a direction for future research.

Comment 4: Have the authors compared their findings to figures from before the pandemic? A quick literature search shows several papers reporting burnout levels in Brazilian anesthesiologists (eg Sousa, Ana Rafaela Campos, and Joana Irene de Barros Mourão. "Burnout em anestesiologia." Revista Brasileira de Anestesiologia 68 (2018): 507-517.)

Response: Thank you for your pertinent question. In our literature review, we identified a systematic review conducted by Sousa et al. (2018), which included 45 articles on burnout in anesthesiology. However, none of these studies specifically examined Brazilian anesthesiologists. Notably, two of the 45 articles were authored by a Brazilian, but these studies were conducted within the North American population, not within Brazil. To the best of our knowledge, the only study that evaluated burnout among Brazilian anesthesiologists during the COVID-19 pandemic focused on anesthesiology residents rather than anesthesiologists of all ages (dos Santos, et al. 2023). This study highlighted the burnout risk among anesthesiology residents during the second wave of COVID-19, providing valuable insights but within a different demographic.

Therefore, a direct comparison with figures from before the pandemic for Brazilian anesthesiologists i

---

## [Decision Letter · Decision Letter 1]

3 Sep 2024

PONE-D-24-02232R1Prevalence of Burnout Syndrome in Brazilian Anesthesiologists during the COVID-19 Pandemic: A Cross-Sectional SurveyPLOS ONE

Dear Dr. Azi,

Thank you for submitting your manuscript to PLOS ONE. After careful consideration, we feel that it has merit but does not fully meet PLOS ONE’s publication criteria as it currently stands. Therefore, we invite you to submit a revised version of the manuscript that addresses the points raised during the review process.

 Please submit your revised manuscript by Oct 18 2024 11:59PM. If you will need more time than this to complete your revisions, please reply to this message or contact the journal office at plosone@plos.org. Please include the following items when submitting your revised manuscript:A rebuttal letter that responds to each point raised by the academic editor and reviewer(s). You should upload this letter as a separate file labeled 'Response to Reviewers'.A marked-up copy of your manuscript that highlights changes made to the original version. You should upload this as a separate file labeled 'Revised Manuscript with Track Changes'.An unmarked version of your revised paper without tracked changes. You should upload this as a separate file labeled 'Manuscript'.If applicable, we recommend that you deposit your laboratory protocols in protocols.io to enhance the reproducibility of your results. Protocols.io assigns your protocol its own identifier (DOI) so that it can be cited independently in the future. For instructions see: https://journals.plos.org/plosone/s/submission-guidelines#loc-laboratory-protocols. Additionally, PLOS ONE offers an option for publishing peer-reviewed Lab Protocol articles, which describe protocols hosted on protocols.io. Read more information on sharing protocols at https://plos.org/protocols?utm_medium=editorial-email&utm_source=authorletters&utm_campaign=protocols.

We look forward to receiving your revised manuscript.

Kind regards,

Francesco Marcatto, Ph.D.

Academic Editor

PLOS ONE

Journal Requirements:

Additional Editor Comments:

Dear Dr. Torres de Araujo Azi,

Thank you for submitting the revised version of your manuscript. I have received positive feedback from a reviewer, but there are still a few points that need to be addressed before your submission can be considered for publication. I encourage you to address all the points raised by the reviewer, paying particular attention to point #3.

Best regards,

Francesco Marcatto

Reviewers' comments:

Reviewer's Responses to Questions

**Comments to the Author**

1. If the authors have adequately addressed your comments raised in a previous round of review and you feel that this manuscript is now acceptable for publication, you may indicate that here to bypass the “Comments to the Author” section, enter your conflict of interest statement in the “Confidential to Editor” section, and submit your "Accept" recommendation.

Reviewer #2: All comments have been addressed

2. Is the manuscript technically sound, and do the data support the conclusions?

Reviewer #2: Yes

3. Has the statistical analysis been performed appropriately and rigorously? 

Reviewer #2: Yes

4. Have the authors made all data underlying the findings in their manuscript fully available?

Reviewer #2: Yes

5. Is the manuscript presented in an intelligible fashion and written in standard English?

Reviewer #2: Yes

6. Review Comments to the Author

Reviewer #2: The rebuttal and text changes are sufficient.

However a few points remain which the authors should consider:

1. Still missing is the connection between the hypothesis in the introduction and the conclusion in the discussion; not closing the circle so to say (see 5). Added to the introduction is now: The hypothesis that burnout prevalence is higher during the pandemic. Higher compared to which numbers? The paper would gain by referencing pre-covid anaesthetist burnout prevalence, preferably from Brasil otherwise from other countries to emphasize the importance of the problem. This is done elsewhere in the paper, but a short sentence in the introduction with references would do. (eg. Something like: Burn out in anesthesiologists is an important problem, international (pre covid) anaesthetist burnout literature report prevalences of … to … %)

2. In your answer to comment 2 you state: “The prevalence of burnout found in our study did not show a significant increase”. Related to above, the question here is: increase compared to which numbers/ what/ when? See 3.

3. In your answer to comment 4 you state: Therefore, a direct comparison with figures from before the pandemic for Brazilian anesthesiologists is not feasible due to the lack of specific pre-pandemic studies within this population. Are you sure there have been no earlier studies into Brazilian anesthetist burnout rates, as you state in your answer to comment 4? How about: Magalhães, Edno, et al. "Prevalence of burnout syndrome among anesthesiologists in the Federal District." Revista brasileira de anestesiologia 65 (2015): 104-110. Coincidentally, the first sentence of this paper closely resembles the first sentence of your paper.

Also: Govêia, Catia Sousa, et al. "Association between burnout syndrome and anxiety in residents and anesthesiologists of the Federal District." Revista brasileira de anestesiologia 68 (2018): 442-446.)

4. Concerning reference 20 consider also referencing this survey from Brasil, in residents anaesthesiology. Pietroski dos Santos, Natanael, et al. "Burnout risk among anesthesiology residents in Brazil during the second wave of COVID-19: a cross-sectional survey." Brazilian Journal of Anesthesiology 73.1 (2022): 120-122.

5. In your conclusion: This study represents a comprehensive assessment of burnout among all Brazilian anesthesiologists during the COVID-19 pandemic, highlighting a high prevalence of burnout and its associate risk factors. This conclusion is in line with your findings.

But: In order to relate to your hypothesis and in line with your findings and discussion, perhaps I would recommend to add: However, burnout rates (in Brazilian anaesthetists) seem not to be higher during the pandemic compared to burnout rates pre-pandemic. (internationally or in Brasil)

I wish my colleagues all the best, hope to meet you sometime at an international conference. Cheers, R.

7. PLOS authors have the option to publish the peer review history of their article (what does this mean?). If published, this will include your full peer review and any attached files.

Reviewer #2: No

---

## [Author Response · Author response to Decision Letter 1]

21 Oct 2024

Prevalence of Burnout Syndrome in Brazilian Anesthesiologists during the COVID-19 Pandemic: A Cross-Sectional Survey

Response to Reviewers – Revised Manuscript (Second Review)

Dear Editor and Reviewer 2,

We sincerely appreciate the thoughtful feedback provided during the review process. Your valuable insights have guided us in refining and strengthening our manuscript. 

In this revised version, we have carefully addressed all the points raised, including adjustments to the introduction and discussion sections, and we have further enhanced the clarity of the conclusion based on your suggestions.

We believe that these revision have significantly improved the overall quality of the work and aligned it more closely with the expectations of PLOS ONE. Thank you again for your constructive input, which has been instrumental in shaping the final version of this manuscript.

We look forward to your review of this revised submission.

Kind regards,

The authors

Reviewer #2 (Highlighted in Yellow)

General Feedback: The rebuttal and text changes are sufficient. However a few points remain which the authors should consider:

Comment 1: Still missing is the connection between the hypothesis in the introduction and the conclusion in the discussion; not closing the circle so to say (see 5). Added to the introduction is now: The hypothesis that burnout prevalence is higher during the pandemic. Higher compared to which numbers? The paper would gain by referencing pre-covid anaesthetist burnout prevalence, preferably from Brasil otherwise from other countries to emphasize the importance of the problem. This is done elsewhere in the paper, but a short sentence in the introduction with references would do. (eg. Something like: Burn out in anesthesiologists is an important problem, international (pre covid) anaesthetist burnout literature report prevalences of ... to ... %)

Response 1: Thank you for your insightful comment. We agree with your suggestion and have revised the introduction to strengthen the connection between the hypothesis and the conclusion. Specifically, we have added references to both pre-pandemic burnout prevalence among Brazilian anesthesiologists and post-pandemic international data, to better contextualize the hypothesis that burnout prevalence increased during the COVID-19 pandemic.

The revised introduction now includes pre-pandemic data from Brazil, where burnout rates ranged from 2.43% to 10.4% in small samples from specific states, alongside international post-pandemic studies reporting burnout rates among anesthesiologists as high as 30% to 60%. This contrast highlights the potential impact of the pandemic on increasing burnout prevalence and provides a stronger foundation for our hypothesis. We believe this revision aligns the introduction more closely with the discussion and conclusions.

Comment 2: In your answer to comment 2 you state: “The prevalence of burnout found in our study did not show a significant increase”. Related to above, the question here is: increase compared to which numbers/ what/ when? See 3.

Response 2: Thank you for this important question. Initially, we based our comparison on a previous study from our research group, "Impact of the COVID-19 Pandemic on the Prevalence of Burnout among Residents in Orthopedics," which was conducted in a small sample of orthopedic residents. This study found no significant increase in burnout prevalence or severity before and during the pandemic. However, after reflecting on your comment and reviewing additional literature, we realize that it is crucial to focus on the specific context of anesthesiology during the pandemic, where burnout has been shown to increase significantly.

In light of this, we revised our approach, incorporating relevant studies on anesthesiologists that highlight the heightened risk and prevalence of burnout during the COVID-19 pandemic. The national and international data, along with our findings, suggest a clear increase in burnout among anesthesiologists, which may not be generalizable from other specialties like orthopedics. We have updated the manuscript to reflect this change in perspective and clarify the increase in burnout specifically within anesthesiology during the pandemic, supported by additional studies.

Comment 3: In your answer to comment 4 you state: Therefore, a direct comparison with figures from before the pandemic for Brazilian anesthesiologists is not feasible due to the lack of specific pre-pandemic studies within this population. Are you sure there have been no earlier studies into Brazilian anesthetist burnout rates, as you state in your answer to comment 4? How about: Magalhães, Edno, et al. "Prevalence of burnout syndrome among anesthesiologists in the Federal District." Revista brasileira de anestesiologia 65 (2015): 104-110. Coincidentally, the first sentence of this paper closely resembles the first sentence of your paper. Also: Govêia, Catia Sousa, et al. "Association between burnout syndrome and anxiety in residents and anesthesiologists of the Federal District." Revista brasileira de anestesiologia 68 (2018): 442-446.)

Response 3: Thank you for bringing these studies to our attention. Initially, we did not include these studies in our work because they focus on specific, localized samples (residents and anesthesiologists in the Federal District), and our study aims to provide a more comprehensive analysis of the situation across Brazil. However, we fully agree that these are the only published pre-pandemic sources on the prevalence of burnout among anesthesiologists in Brazil, and it is essential to include and compare them with the broader national context during the COVID-19 pandemic.

We have now incorporated the studies by Magalhães et al. (2015) and Govêia et al. (2018) into both the introduction and discussion sections of the manuscript. These studies, though based on small and localized samples, provide valuable pre-pandemic data on Brazilian anesthesiologists, with burnout prevalence ranging from 2.43% to 10.4%. These findings allow us to make more meaningful comparisons with our post-pandemic data, which shows a marked increase in burnout prevalence, and thus strengthens the conclusions of our study.

Comment 4: Concerning reference 20 consider also referencing this survey from Brasil, in residents anaesthesiology. Pietroski dos Santos, Natanael, et al. "Burnout risk among anesthesiology residents in Brazil during the second wave of COVID-19: a cross-sectional survey." Brazilian Journal of Anesthesiology 73.1 (2022): 120-122.

Response 4: Thank you for recommending this relevant reference. We have added the study by Pietroski dos Santos et al. (2022) to our manuscript. This study provides critical insight into the burnout risk among anesthesiology residents during the second wave of COVID-19, with findings showing that 73.2% of residents were at risk of burnout and 57.1% were at high risk.

Although this study used a different instrument (the Oldenburg Burnout Inventory, rather than the Maslach Burnout Inventory), it still offers valuable data on the increasing burnout risk during the pandemic. We have noted the methodological differences between our study and this reference in the discussion section, emphasizing that despite the different measurement tools, the consistent findings of increased burnout risk in both studies underline the severity of the issue among anesthesiology professionals during the pandemic.

Comment 5: In your conclusion: This study represents a comprehensive assessment of burnout among all Brazilian anesthesiologists during the COVID-19 pandemic, highlighting a high prevalence of burnout and its associate risk factors. This conclusion is in line with your findings. But: In order to relate to your hypothesis and in line with your findings and discussion, perhaps I would recommend to add: However, burnout rates (in Brazilian anaesthetists) seem not to be higher during the pandemic compared to burnout rates pre-pandemic. (internationally or in Brasil.

Response 5: Thank you for your insightful suggestion. We agree that incorporating a comparison between pre-pandemic and pandemic burnout rates will provide a more nuanced understanding of the findings. While we acknowledge the elevated burnout rates observed during the pandemic, the available pre-pandemic studies in Brazil show similarly high levels of burnout, suggesting that the pandemic may not have led to a significant increase in burnout rates among anesthesiologists. We have revised the conclusion accordingly to reflect this point and align it with our discussion and findings.

---

## [Editor Report · Decision Letter 2]

28 Oct 2024

Prevalence of Burnout Syndrome in Brazilian Anesthesiologists during the COVID-19 Pandemic: A Cross-Sectional Survey

PONE-D-24-02232R2

Dear Dr. Azi,

We’re pleased to inform you that your manuscript has been judged scientifically suitable for publication and will be formally accepted for publication once it meets all outstanding technical requirements.

Kind regards,

Francesco Marcatto, Ph.D.

Academic Editor

PLOS ONE
---

## [Editor Report · Acceptance letter]

8 Jan 2025

PONE-D-24-02232R2 

PLOS ONE

Dear Dr. Azi, 

I'm pleased to inform you that your manuscript has been deemed suitable for publication in PLOS ONE. Congratulations! Your manuscript is now being handed over to our production team.

Kind regards, 

on behalf of

Dr. Francesco Marcatto 

Academic Editor

PLOS ONE